# Comparison of COVID-19 Pandemic-Related Stress among Frontline Medical Personnel in Daegu City, Korea

**DOI:** 10.3390/medicina57060583

**Published:** 2021-06-07

**Authors:** Su-Jeong Shin, Yun-Jeong Kim, Hyun-Wook Ryoo, Sungbae Moon, Sang-Hun Lee, You-Ho Moon, Tae-Chang Jang, Dong-Chan Park

**Affiliations:** 1Department of Emergency Medicine, Yeungnam University Medical Center, Daegu 42415, Korea; dongle7979@gmail.com; 2Department of Emergency Medicine, School of Medicine, Kyungpook National University, Daegu 41944, Korea; ryoo@knu.ac.kr (H.-W.R.); snbaem@gmail.com (S.M.); 3Department of Emergency Medicine, Dongsan Medical Center, Keimyung University, Daegu 42601, Korea; saorang7@naver.com; 4Department of Emergency Medicine, Yeungnam University College of Medicine, Daegu 42415, Korea; zezec@naver.com; 5Department of Emergency Medicine, School of Medicine, Daegu Catholic University, Daegu 42472, Korea; bizkit96@naver.com; 6Department of Emergency Medicine, Daegu Fatima Hospital, Daegu 41199, Korea; dongchan00@hanmail.net

**Keywords:** mental health, medical staff, COVID-19 pandemic, stress anxiety

## Abstract

*Background and Objectives*: Frontline medical staff usually experience high levels of stress, which could greatly impact their work output. We conducted a survey to investigate the level of stress and its association with job types, work departments, and medical centers among COVID-19 pandemic frontline medical personnel. *Materials and Methods*: We conducted a cross-sectional survey using a self-administered questionnaire among 307 frontline medical staff who cared for COVID-19 patients in Daegu city. We used a 33-item questionnaire to assess respondents’ general characteristics, job stress, personal effects associated with the COVID-19 pandemic, and their stress level. A general health questionnaire-12 (GHQ-12) was included in our questionnaire. *Results*: Majority (74.3%) of the respondents were in the stress group. The mean GHQ-12 score was 14.31 ± 4.96. More females (67.4%, *p* < 0.05) and nurses (73.3%, *p* = 0.001) were in the stress group compared to males and doctors. Medical staff in the general ward considered the severity of the COVID-19 pandemic situation higher. Nurses perceived work changes (*p* < 0.05), work burden (*p* < 0.05), and personal impact (*p* < 0.05) more serious than doctors. Medical staff in Level 3 emergency department (ED) perceived a lack of real-time information (*p* = 0.012), a lack of resources, and negative personal impacts associated with the pandemic as more serious than staff in Level 1 and Level 2 EDs. Medical staff in the intensive care unit perceived work changes (*p* < 0.05), work burden (*p* < 0.05), and lack of personal protective equipment (*p* = 0.002) as more serious than staff in the ED and general ward. *Conclusion*: Providing real-time information and resources for reducing work burden and negative personal impact is central to maximizing the work output of the COVID-19 pandemic frontline medical staff. Supporting their mental health through regular programs and intervention is also imperative.

## 1. Introduction

The COVID-19 outbreak started in December 2019 in Wuhan, China. The World Health Organization declared the outbreak a Public Health Emergency of International Concern on January 30, 2020, and a pandemic on March 11 of the same year. To date, there are 132,046,206 confirmed cases and 2,867,242 deaths due to COVID-19 worldwide [1].

Since the first case of COVID-19 was confirmed in South Korea on 19 January 2020, a surge in the number of cases was reported by 18 February 2020 in the form of cluster infection related to the Shincheonji religious group in Daegu city, a major epidemic area of South Korea’s COVID-19 outbreak [2,3]. Even though Korea is high on the Global Health Security index (the most prepared country for epidemic preparedness) [4], after that surge, many emergency departments (EDs) were repeatedly closed down, with 40 temporary ED closures in Daegu. The median and total duration of temporary ED closures for Levels 1 and 2 EDs were 17.5 h and 769 h, respectively between 18 February and 25 March 2020 [5]. Furthermore, medical staff on duty were quarantined as part of every ED closure, and for treatment of COVID-19-confirmed patients, additional, untrained medical staff were reassigned to the ED, general ward (GW), and intensive care unit (ICU). Before the pandemic, in Korea, especially in Daegu city, there were few infectious patients who needed to be quarantined in the ED, so usually, medical staff practiced without personal protective equipment (PPE). After the pandemic, the unknown infectious disease forced frontline medical staff to wear PPE. However, nobody could guarantee the medical staff’s safety. So, medical staff in Daegu city, who were not ready to respond at that time, were in chaos and lived in constant fear of infection every day.

Medical staff working on the frontline, especially in the ED, usually experience high levels of stress, both physical and mental [6,7]. As many studies have shown that the psychiatric and post-traumatic morbidity of frontline medical staff has increased since the severe acute respiratory syndrome (SARS) outbreak, the COVID-19 pandemic era might have increased the fear of the unknown due to it being a novel, highly contagious, and sometimes fatal disease [8]. Therefore, our study aimed to evaluate the level and kind of stress and its relationship with job types, work departments, and medical centers among COVID-19 pandemic frontline medical staff.

## 2. Materials and Methods

### 2.1. Study Design, Participants, and Procedure

Daegu is a metropolitan city with a population of about 2.4 million people (as of 2020). It has 16 emergency medical centers which had been designated as emergency medical centers by the government (2 Level 1 EDs, 4 Level 2 EDs, and 10 Level 3 EDs). The number of ED medical staff is 514 (149, 214, and 151 in each). The number of frontline medical staff who care for COVID-19 patients in the GW and the ICU is 150 (as of March 2020, doctors 18, nurses 132). We conducted a cross-sectional survey using a self-administered questionnaire for 664 frontline medical staff who worked in the ED, GW, and ICU. Data was collected between 18 May and 3 June 2020 (Figure 1). Among them, 307 medical staff (82 doctors and 225 nurses) responded. The questionnaire was electronic-based (Google survey) and was deployed using a smartphone. Informed consent was obtained from all the participants prior to participation and they received monetary compensation for their time and data.

The questionnaire consisted of 33 questions which were divided into three parts, general characteristics, job stress and personal stress associated with the COVID-19 pandemic since February 2020, and general health questionnaire-12 (GHQ-12). All items were measured on a 5-point Likert scale (“strongly disagree”, “disagree”, “neither agree nor disagree”, “agree”, “strongly agree”) except GHQ-12 which was measured on a 4-point Likert scale (“never”, “rarely”, “often”, “every time”). The section on job stress and personal stress associated with the COVID-19 pandemic was further divided into five parts: General, changes in work, work burden, lack of resources, and personal effect.

### 2.2. Outcome Definition

Using the GHQ-12, general health was measured on a 36-point scale, with each question assigned a score between 0 and 3, depending on the participant’s response. Bimodal distribution (0-0-1-1) of the GHQ-12 data was used to classify the participants into two groups, the stress group and the no-stress group. The cut-off score for the bimodal method was set at 2/3 while that of the Likert scale was set at 11/12, determined by previous studies that have reported Likert scores [9,10,11,12]. The higher the GHQ-12 score, the higher the level of stress.

### 2.3. Data Analysis

Data were analyzed using IBM SPSS version 26.0 (IBM Corp., Armonk, NY, USA). Frequencies and percentages were generated for nominal variables and means and standard deviations for continuous variables. The chi-square test was performed to examine the difference between the stress group and the no-stress group. The *t*-test and ANOVA were used to investigate the factors that differ between groups. For continuous variables, comparisons between three groups were performed using ANOVA, and detailed comparisons were performed through *post-hoc* analysis. Statistical significance was set at a *p*-value < 0.05. Exploratory factor analysis (EFA) was carried out to determine the factorial validity of our questionnaire. Prior to the extraction of factors, the Kaiser-Meyer-Olkin (KMO) measure of sampling adequacy was checked to evaluate the fitness of the data for factor analysis. Initially, principal component analysis (PCA) with varimax rotation was used to extract factors based on multiple criteria, including an Eigenvalue >1, the Scree test, a factor loading coefficient > 0.4, and the cumulative percent of variance extracted. Finally, parallel analysis was carried out to determine the ultimate factors to be related. The internal consistency reliability of each factorially derived scale was assessed by calculating Cronbach’s alpha.

### 2.4. Ethics Statement

The study protocol was approved by the Institutional Review Board of Kyungpook National University Chilgok Hospital (IRB No. 2020-06-018). Informed consent was confirmed by the IRB.

## 3. Results

### 3.1. Validation

The KMO measure of sampling adequacy value was 0.848, exceeding the recommended value of 0.6, and Bartlett’s test of sphericity reached statistical significance (*p* = 0.000), supporting factorability of the correlation matrix. PCA with Varimax rotation demonstrated the presence of five components with Eigenvalues exceeding one. An inspection of the Scree plot also revealed a clear break after the 5th component. Using this result, it was decided to retain five component solutions, which explained a total of 62.8% of the variance. We defined these five components as ‘General’, ‘Changes in Work’, ‘Work Burden’, ‘Lack of Resources’, and ‘Personal Effect’ respectively. The values of Cronbach’s alpha were 0.655, 0.718, 0.766, 0.670, and 0.662 respectively.

### 3.2. General Characteristics

Based on the bimodal method of calculating the GHQ-12 score, participants were categorized into two groups: a stress group and a no-stress group (Table 1). Of the 307 respondents, 228 (74.3%) belonged to the stress group while 79 (25.7%) belonged to the no-stress group. Women (82.6%) were more likely to belong to the stress group than men (57.0%).

The mean age of participants in the stress and no-stress groups was 33.35 ± 8.94 years and 34.19 ± 8.51 years respectively, with the stress group 0.84 years younger on average than the no-stress group. The difference in the GHQ-12 score between the no-stress group (8.90 ± 2.71) and the stress group (16.19 ± 4.41) was 7.29 points. According to job type, the proportion of nurses (80.0%) in the stress group was significantly higher than the specialist doctors (60.8%) and resident doctors (54.8%). In terms of information acquisition on the COVID-19 pandemic, 63.4% acquired information through the internet and TV news, 25.6% through the government’s daily briefing, and 11.0% through a hospital bulletin. There was, however, no significant difference in the source of information acquisition between the two groups.

### 3.3. Comparison of GHQ-12 Scores between Subgroups

The average GHQ-12 score by gender, marital status, offspring, elderly family member, underlying disease, job type, ED classification, and work department was 12 points or higher (Table 2). The difference in the GHQ-12 score by gender was statistically significant, with an average of 2.58 points higher among the females (15.15 ± 4.64) compared to the males (12.57 ± 5.17). Nurses (14.69 ± 4.67) had the highest GHQ-12 score, followed by specialist doctors (13.47 ± 5.84), and residents (12.97 ± 5.24). The higher the ED level, the higher the GHQ-12 score (13.95 ± 4.80 vs. 14.22 ± 5.00 vs. 15.72 ± 5.10, respectively for Levels 1, 2, and 3 ED). ICU staff (16.05 ± 4.42) had the highest GHQ-12 score followed by the GW (15.57 ± 6.07) and the ED (14.06 ± 4.87). While there was a significant difference in the GHQ-12 score by gender, there was no significant difference by marital status, offspring, elderly family member, underlying disease job type, ED classification, or work department.

### 3.4. Comparison of GHQ-12 Scores between Stress Group and No-Stress Group

A total of 21 questions, which comprised questions on general considerations (4 items), work changes (3 items), work burden (6 items), lack of resources (3 items), and personal effect (5 items) were used to obtain information on the content of stress (Table 3). The average 5-point Likert score for each item was used to classify the participants into stress group and no-stress group, and thereafter compared and analyzed. There was a significant difference in the perception that the COVID-19 pandemic is influential enough to change the emergency medical service (EMS) system in the future between the no-stress group (4.24 ± 0.74) and stress group (4.46 ± 0.65). With regards to work changes, there was a significant difference in working time, work intensity, and working contents between the two groups. In terms of work burden, significantly more participants in the stress group reported increased fatigue due to wearing PPE at work, different patient treatment processes, frequent changes, or absent guidelines for treating febrile and respiratory disease patients, and increased risk of infection by contact with COVID-19 confirmed patients. Significantly, more participants in the stress group reported a lack of PPE. Regarding the personal influence questions, significantly more participants in the stress group were worried about COVID-19 symptoms, the impact of their job on their family, being a threat to their family’s safety, and the negative effect of their job on their life.

### 3.5. Comparison of Likert Score According to Job Type (Doctors vs. Nurses)

The majority (80.0%) of the nurses and more than half (58.5%) of the doctors were in the stress group category. Doctors had a significantly higher mean age (38.50 ± 8.09 years) compared to the nurses while the nurses (14.69 ± 4.67) had a significantly higher GHQ-12 score than doctors (13.28 ± 5.59).

The differences between job types (doctor group and nurse group) were investigated for the 21 questions of the same content described above (Table 4). The average 5-point Likert score for all items was higher than 3 points. In all three questions about work changes (working time, work intensity, and working content changes), the difference between job type and work changes was statistically significant. With regards to factors related to work burden, more nurses compared to the doctors, significantly reported increased fatigue of wearing PPE at work, different patient treatment processes, frequent changes or absent guidelines for treating febrile and respiratory disease patients, and increased risk of infection by contact with COVID-19 confirmed patients. Concerning the lack of resources, there was a significant difference in lack of beds for febrile, respiratory patients, lack of PPE, and lack of medical staff and assistants between the job types. Significantly, more doctors reported a lack of beds for febrile, respiratory patients while more nurses significantly reported a lack of PPE and lack of medical staff and assistants. More nurses, compared to doctors, significantly expressed concerns about efforts to maintain a healthy lifestyle after COVID-19, COVID-19 symptoms while working, the effect of their job on their family life, and being a threat to their family’s safety.

### 3.6. Comparison of Likert Score According to ED Classification

Table 5 reports the comparison of Likert scores for the 21 questions according to ED classification. The average 5-point Likert score for all items was higher than 3 points. In all three questions about work changes, scores were not significantly different between each ED. However, for questions on work intensity and working contents change, the average score of all EDs was higher than 4.2 points.

There were no statistically significant differences between the EDs and all six work burden constructs. However, it was found that Level 2 and Level 3 EDs complained of more fatigue from wearing PPE than Level 1 ED. The frequent changes or absence of guidelines for treating febrile and respiratory patients score was higher at Level 1 and Level 2 EDs than Level 3 ED. In addition, awareness of the risk of COVID-19 infection was found to be greater at Level 3 ED than at Level 1 ED or Level 2 ED.

Although there was no significant difference between the ED scores and the lack of resources, all three EDs recognized that resources were insufficient. However, Level 3 ED was more aware of the insufficiency of resources than Level 1 ED and Level 2 ED. The perception that there is a lack of beds and manpower for patients was higher than the PPE shortage. The scores for anxiety about symptoms of COVID-19, the personal, familial, and social impacts of COVID-19 pandemic on frontline health care workers, and being a threat to the safety of their own family, were higher in Level 3 ED than the other two EDs.

### 3.7. Comparison of Likert Score According to Work Department

We investigated the differences in Likert scores according to the work department (Table 6). The difference between the work department and perceptions that (1) the COVID-19 pandemic experience will help to respond well to new infectious diseases in the future and (2) the pandemic is a serious situation that could change the EMS system was statistically significant. Participants working in the ICU had a significantly higher score for point 1 above than the other departments while those working in the GW had a significantly higher score for point 2 above than those in the ED.

There was a significant difference in the working time and work intensity between the departments, with ICU staff significantly scoring the highest among the three departments. More ICU staff significantly reported that increased fatigue due to wearing PPE, delay in patient treatment due to the COVID-19 pandemic, and risk of infection because of contact with COVID-19 positive patients constituted a work burden. Similarly, more ICU staff reported a PPE shortage compared to the GW and ED staff, and this difference was statistically significant. Regarding personal effect, there was no statistically significant difference in the scores between the three departments. However, *post-hoc* analysis showed that the score in the ED was significantly lower than that of the ICU.

## 4. Discussion

Frontline medical personnel usually experience a high level of job and social stress [13]. Studies during the SARS epidemic showed that 29 to 35 percent of frontline medical personnel experienced high stress, and the more confirmed cases they contacted, the more stress they encountered [14]. Additionally, studies during the middle east respiratory syndrome (MERS) epidemic reported that many frontline medical staff suffered from job stress, post-traumatic stress disorder (PTSD), anxiety, fear, and depression [15,16]. Findings from these studies thus suggest that COVID-19 infection, which has been prevalent in a wider area and for a longer period than the SARS and MERS outbreaks, will have more stress and psychological effects on frontline medical personnel.

Therefore, in our study, we investigated the proportion of COVID-19 frontline medical staff who are under stress, the reasons for the stress, and the differences by job type, ED classification, and work department. When we categorized participants into a stress group and no-stress group based on the GHQ-12 score, the stress group accounted for 74.3% of the total. The overall average of the GHQ-12 score was 14.31 while the stress group’s average score was 16.19. These scores were 3.31 and 5.19 higher than the cut-off value of 11 respectively. The above results indicate that a good number of COVID-19 frontline medical personnel are under stress. Particularly, more women and nurses were in the stress group and had higher GHQ-12 scores compared to the men and doctors. This suggests that gender and job type have a strong influence on stress experienced by these frontline health workers. This may not be unconnected to gender roles since the majority of the nurses in this study were also females.

Factors responsible for the stress were studied and compared between the stress and no-stress groups. Between these groups, the stress group perceived that the COVID-19 pandemic situation is remarkably serious and they experienced more personal and work-related stress. In many of the domains of stress studied (almost all areas of work changes and burden, lack of resources, and personal influence), the level of stress was higher among the nurses, which parallels previous studies [17,18]. Studies have shown that even when not in pandemic situations, nurses working in the ED encounter high levels of fatigue and stress, and this has a great effect on self-efficacy [6]. Many studies in the COVID-19 pandemic era, as well as the SARS epidemic era, have also indicated that nurses experience high levels of stress including those that are work-related, health concerns, and social isolation, and thus require intervention [19,20,21]. These compare well with the results of our study. A plausible explanation for the high level of stress experienced by the nurses in this study is that in the course of patients’ treatment, nurses had more contact with COVID-19 confirmed cases than doctors. This may have made them become more sensitive to work changes and to consider their work more burdensome.

In the analysis according to the ED classification, Level 3 ED scores for real-time hospital bed status information, lack of protective equipment, the experience of COVID-19-like symptoms during work, and the negative perception of frontline medical staff were significantly higher than in Levels 1 and 2 EDs. Unlike Level 1 ED that treats more COVID-19 confirmed patients, Level 3 ED scored higher in anxiety and negative perceptions, despite treating fewer patients with confirmed or suspected infectious diseases. This result shows that the level of anxiety and work stress experienced by medical staff who treat COVID-19 confirmed patients in Level 1 ED is less than those of Level 3 ED. The reason for this finding could be because Level 1 ED received more support through increased human resources and better real-time information than Level 3 ED. These results are in line with the results of previous studies that sufficient resources, fast and accurate information, and timely precautions can reduce the anxiety and depression of medical staff [20].

Medical staff in the GW perceived the severity of the pandemic situation more seriously than those in the ED. GW staff have less experience treating infectious disease patients than those in the ED or ICU, so the perceived severity of the pandemic would expectedly be greater. The extension of working hours and the increase in work intensity were observed in all departments, and the average score for questions related to work burden exceeded 4 points in all. Fatigue due to wearing PPE, delay in patient treatment, and risk of infection due to treatment of confirmed patients were observed across the working departments. More ICU staff reported a lack of PPE and negative effect on personal daily life from being frontline medical personnel compared to those in the ED. On account of their work department, ICU staff encounter patients with more severe COVID-19 infection, and this may have informed their perception.

Many studies have shown that the general public, as well as frontline medical staff, have suffered from depression, anxiety, and distress due to the COVID-19 pandemic [21,22]. In addition, according to previous studies, healthcare workers experience severe emotional stress during the outbreak of novel infectious diseases (e.g., severe acute respiratory syndrome, Ebola virus disease), and healthcare workers also experience burnout, PTSD, depression, and anxiety after the outbreak [16,23]. The findings of this study have been shown to be consistent with the above studies and thus underscore the need for the mental health of frontline medical staff to take center stage. Because their risks for anxiety and fatigue stem from their role as frontline medical staff, their mental health needs to be monitored and supported periodically during the pandemic period as well as afterward [24].

In our study, it was found that the degree of distress was significantly higher than in previous studies. This could be because COVID-19, a novel infectious disease, spread quickly and fatally within the study area, which had no previous experience with new infectious diseases (SARS and MERS). Even though the general public and frontline medical staff are gradually being vaccinated, the risk of spread still remains high due to the uncertain spread and variant strains of COVID-19 [25].

Due to the foregoing, it is important for multidisciplinary mental health teams established by health authorities at regional and national levels (including psychiatrists, psychiatric nurses, clinical psychologists, and other mental health workers) to deliver mental health support to health workers [22,26,27,28,29]. Furthermore, they can serve as the foundation in readiness for new infectious disease epidemics in the future [30,31].

Our study has some limitations. First, there are limitations to generalize our results, as our study was limited to a local area (Daegu city) and we used a non-standardized questionnaire in our survey. It can be reinforced by a nationwide survey and by using a standardized questionnaire in follow-up studies. Second, there is a lack of a follow-up study. Our study was conducted after the first surge of COVID-19 patients. So, a follow-up study is necessary to confirm our first result and to evaluate the change.

## 5. Conclusions

Whether in a pandemic situation or not, it is necessary to monitor healthcare workers’ stress and psychologic problem, and it is also needed that preparing regular programs to support medical staff’s mental health. Providing real-time information and resources for reducing work burden and negative personal impact is also necessary for protecting our frontline medical staff.

## Figures and Tables

**Figure 1 medicina-57-00583-f001:**
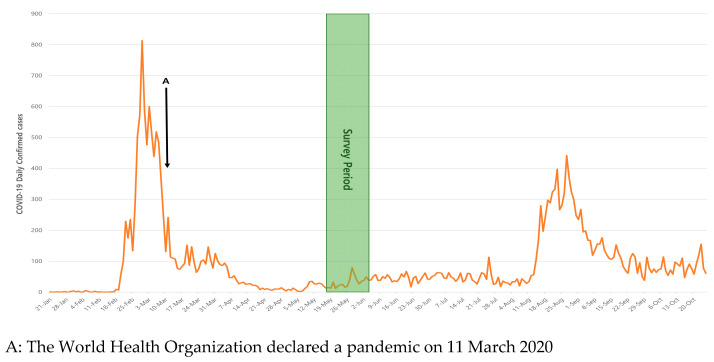
Study Period.

**Table 1 medicina-57-00583-t001:** General Characteristics of Respondents.

	No Stress Group (*n* = 79, 25.7%)	Stress Group (*n* = 228, 74.3%)	*p*
Gender, *n* (%)			0.000 *
Male, *n* = 100	43 (43.0)	57 (57.0)	
Female, *n* = 207	36 (17.4)	171 (82.6)	
Age (mean ± SD)	34.19 ± 8.51	33.35 ± 8.94	0.455
Sum of GHQ ^†^-12 (range 0 ~ 36, mean ± SD)	8.90 ± 2.71	16.19 ± 4.11	0.000 *
Marital status, *n* (%)			0.795
Single	50 (25.3)	148 (74.7)	
Married	29 (26.6)	80 (73.4)	
Offspring, *n* (%)			0.546
No	54 (24.8)	164 (75.2)	
Yes	25 (28.1)	64 (71.9)	
Elderly family member, *n* (%)			0.481
No	71 (26.4)	198 (73.6)	
Yes	8 (21.1)	30 (78.9)	
Underlying disease, *n* (%)			0.585
No, I do not have	73 (26.2)	206 (73.8)	
Yes, I have	6 (21.4)	22 (78.6)	
Job type, *n* (%)			0.001 *
Doctor (Specialist)	20 (39.2)	31 (60.8)	
Doctor (Resident)	14 (45.2)	17 (54.8)	
Nurse	45 (20.0)	180 (80.0)	
Working year (mean ± SD)			0.000 *
Specialist	13.45 ± 6.05	11.42 ± 6.63	0.266
Nurse	7.49 ± 8.50	8.88 ± 9.05	0.352
Emergency center classification, *n* (%)			0.224
Level 1	25 (27.2)	67 (72.8)	
Level 2	49 (27.4)	130 (72.6)	
Level 3	5 (13.9)	31 (86.1)	
Work department, *n* (%)			0.461
ED ^‡^	70 (26.6)	193 (73.4)	
ICU ^§^	3 (14.3)	18 (85.7)	
General Ward	6 (26.1)	17 (73.9)	
Disaster training in 2 years, *n* (%)			0.681
No	36 (24.7)	110 (75.3)	
Yes	43 (26.7)	118(73.3)	
Source of information about COVID-19, *n* (%)			0.566
Internet, TV news	50 (25.8)	144 (74.2)	
KDCA ^∥^ daily briefing	18 (22.8)	61 (77.2)	
Hospital bulletin	11 (32.4)	23 (67.6)	

* *p* < 0.05, ^†^ GHQ; General Health Questionnaire, ^‡^ ED; Emergency Department, ^§^ ICU; Intensive Care Unit, ^∥^ KDCA; Korea Disease Control and Prevention Agency.

**Table 2 medicina-57-00583-t002:** Comparison of General Health Questionnaire (GHQ)-12 scores between subgroups.

	*n* (%)	GHQ ^†^-12 (Mean ± SD)	*p*
Total	307	14.31 ± 4.96	
Gender			0.000 *
Male	100 (32.6)	12.57 ± 5.17	
Female	207 (67.4)	15.15 ± 4.64	
Marital Status			0.580
Single	198 (64.5)	14.43 ± 4.95	
Married	109 (35.5)	14.10 ± 5.01	
Offspring			0.290
No	218 (71.0)	14.50 ± 4.99	
Yes	89 (29.0)	13.84 ± 4.89	
Elderly family member			0.886
No	269 (87.6)	14.30 ± 5.00	
Yes	38 (12.4)	14.42 ± 4.76	
Underlying disease			0.654
No, I do not have	279 (90.9)	14.27 ± 4.96	
Yes, I have	28 (9.1)	14.71 ± 5.11	
Job type			0.080
Doctor (Specialist)	51 (16.6)	13.47 ± 5.84	
Doctor (Resident)	31 (10.1)	12.97 ± 5.24	
Nurse	225 (73.3)	14.69 ± 4.67	
Emergency center classification			0.176
Level 1	92 (30.0)	13.95 ± 4.80	
Level 2	179 (58.3)	14.22 ± 5.00	
Level 3	36 (11.7)	15.72 ± 5.10	
Working place			
ED ^‡^	263 (85.7)	14.06 ± 4.87	0.096
ICU ^§^	21 (6.8)	16.05 ± 4.42	
General Ward	23 (7.5)	15.57 ± 6.07	

* *p* < 0.05, ^†^ GHQ; General Health Questionnaire, ^‡^ ED; Emergency Department, ^§^ ICU; Intensive Care Unit.

**Table 3 medicina-57-00583-t003:** Comparison of General Health Questionnaire-12 Scores between Stress group and No-stress group.

	Likert Scale (Mean ± SD)			*p*
	No-Stress Group (*n* = 79)	Stress Group (*n* = 228)	Total (*n* = 309)	
**General**				
The experience of COVID-19 will enable us to cope well with new infectious diseases in the future.	4.08 ± 0.78	3.89 ± 0.83	3.94 ± 0.82	0.083
COVID-19 pandemic is serious enough to change the emergency medical service system in the future.	4.24 ± 0.74	4.46 ± 0.65	4.40 ± 0.68	0.014 *
The information of each hospital’s bed status or closure is available in real-time.	3.34 ± 0.93	3.3 ± 1.063	3.33 ± 1.03	0.924
The disaster training is helpful in the COVID-19 pandemic response.	3.21 ± 0.88	3.19 ± 1.12	3.19 ± 1.06	0.894
**Changes in Work**				
The duration of my work has got longer since the COVID-19 pandemic.	3.32 ± 1.04	3.77 ± 1.06	3.65 ± 1.07	0.001 *
The intensity of my work has changed since the COVID-19 pandemic.	4.09 ± 0.77	4.38 ± 0.80	4.30 ± 0.80	0.005 *
My work has changed since the COVID-19 pandemic.	4.11 ± 0.68	4.38 ± 0.80	4.31 ± 0.74	0.004 *
**Work Burden**				
The long time to confirm the result of the COVID-19 PCR test.	4.28 ± 0.78	4.46 ± 0.75	4.41 ± 0.76	0.075
Increased fatigue due to wearing protective clothing during work.	4.28 ± 0.78	4.54 ± 0.67	4.48 ± 0.71	0.004 *
Delay in patient treatment associated with the COVID-19 pandemic.	4.30 ± 0.74	4.46 ± 0.67	4.42 ± 0.69	0.090
Treatment process is different from usual, and the working process is not familiar.	3.86 ± 0.76	4.25 ± 0.78	4.15 ± 0.79	0.000 *
Febrile, respiratory patient treatment guidelines are absent or frequently changed.	4.13 ± 0.71	4.33 ± 0.73	4.28 ± 0.73	0.034 *
Risk of infection from contact with confirmed COVID-19 patients.	4.13 ± 0.69	4.53 ± 0.65	4.43 ± 0.68	0.000 *
**Lack of Resources**				
Lack of beds for febrile, respiratory patients.	4.41 ± 0.67	4.45 ± 6.01	4.44 ± 0.62	0.565
Lack of personal protective equipment.	3.47 ± 0.90	3.96 ± 1.00	3.83 ± 1.00	0.000 *
Lack of medical staff and assistants due to the operation of the COVID-19 care unit.	4.39 ± 0.67	4.50 ± 0.73	4.47 ± 0.72	0.182
**Personal Effect**				
I make more effort to maintain a healthy lifestyle after the COVID-19 pandemic.	3.25 ± 1.03	3.47 ± 1.14	3.41 ± 1.11	0.137
I used to be worried about COVID-19-like symptoms during work.	3.18 ± 1.22	3.79 ± 1.10	3.63 ± 1.16	0.000 *
Working as frontline medical personnel around infectious patients has affected my family life.	3.75 ± 0.98	4.23 ± 0.82	4.10 ± 0.89	0.000 *
I think I could be a threat to the safety of my family.	4.14 ± 0.69	4.54 ± 0.61	4.43 ± 0.66	0.000 *
Being frontline medical personnel around infectious patients has negatively affected my social and daily life.	3.00 ± 1.05	3.63 ± 1.11	3.47 ± 1.13	0.000 *

* *p* < 0.05, All items were measured on a 5-point Likert scale (“1-strongly disagree”, “2-disagree”, “3-neither agree nor disagree”, “4-agree”, “5-strongly agree”).

**Table 4 medicina-57-00583-t004:** Comparison of Likert Score According to Job Type (Doctors vs. Nurses).

	Likert Scale (Mean ± SD)		*p*
	Doctors (*n* = 82)	Nurses (*n* = 225)	
**General**			
The experience of COVID-19 will enable us to cope well with new infectious diseases in the future.	3.99 ± 0.75	3.92 ± 0.85	0.498
COVID-19 pandemic is serious enough to change the emergency medical service system in the future.	4.49 ± 0.63	4.37 ± 0.68	0.173
The information of each hospital’s bed status or closure is available in real-time.	3.43 ± 1.05	3.30 ± 1.02	0.332
The disaster training is helpful in the COVID-19 pandemic response	2.95 ± 1.09	3.28 ± 1.05	0.093
**Changes in Work. Since the Covid-19 pandemic:**			
My work time has increased	3.37 ± 1.04	3.76 ± 1.07	0.005 *
My work intensity has changed	3.90 ± 0.84	4.45 ± 0.73	0.000 *
My work content has changed	4.10 ± 0.71	4.39 ± 0.74	0.002 *
**Work Burden**			
The long time to confirm the result of the COVID-19 PCR test	4.41 ± 0.75	4.41 ± 0.77	0.954
Increased fatigue due to wearing protective clothing during work	4.12 ± 0.79	4.60 ± 0.63	0.000 *
Delay in patient treatment associated with the COVID-19 pandemic	4.46 ± 0.67	4.40 ± 0.69	0.475
Treatment process is different from usual, and the working process is not familiar.	3.78 ± 0.85	4.28 ± 0.73	0.000 *
Febrile, respiratory patient treatment guidelines are absent or frequently changed.	4.02 ± 0.80	4.37 ± 0.68	0.000 *
Risk of infection from contact with confirmed COVID-19 patients	4.12 ± 0.73	4.54 ± 0.63	0.000 *
**Lack of Resources**			
Lack of beds for febrile, respiratory patients	4.59 ± 0.57	4.39 ± 0.63	0.013 *
Lack of personal protective equipment	3.33 ± 0.97	4.02 ± 0.95	0.000 *
Lack of medical staff and assistants due to the operation of the COVID-19 care unit.	4.20 ± 0.74	4.57 ± 0.68	0.000 *
**Personal Effect**			
I make more effort to maintain a healthy lifestyle after the COVID-19 pandemic.	3.06 ± 1.06	3.54 ± 1.11	0.001 *
I used to be worried about COVID-19-like symptoms during work.	3.34 ± 1.10	3.74 ± 1.16	0.008 *
Working as frontline medical personnel around infectious patients has affected my family life.	3.93 ± 0.86	4.17 ± 0.86	0.035 *
I think I could be a threat to the safety of my family	4.16 ± 0.79	4.53 ± 0.57	0.000 *
Being frontline medical personnel around infectious patients has negatively affected my social and daily life.	3.34 ± 1.17	3.52 ± 1.11	0.232

* *p* < 0.05, All items were measured on a 5-point Likert scale (“1-strongly disagree”, “2-disagree”, “3-neither agree nor disagree”, “4-agree”, “5-strongly agree”).

**Table 5 medicina-57-00583-t005:** Comparison of Likert Score According to Emergency Department Classification.

	Likert Scale (Mean ± SD)			*p*	
	Level 1 (*n* = 92)	Level 2 (*n* = 179)	Level 3 (*n* = 36)		
**General**					
The experience of COVID-19 will enable us to cope well with new infectious diseases in the future.	4.00 ± 0.76	3.89 ± 0.86	4.03 ± 0.77	0.243	
COVID-19 pandemic is serious enough to change the emergency medical service system in the future.	4.45 ± 0.64	4.35 ± 0.70	4.53 ± 0.65	0.810	
The information of each hospital’s bed status or closure is available in real-time.	3.58 ± 0.83	3.23 ± 1.06	3.19 ± 1.24	0.012 *	1 > 3
The disaster training is helpful in the COVID-19 pandemic response	3.37 ± 1.09	3.07 ± 1.06	3.60 ± 0.91	0.465	
**Changes in Work**					
The duration of my work has got longer since the COVID-19 pandemic.	3.63 ± 1.08	3.65 ± 1.06	3.72 ± 1.16	0.445	
The intensity of my work has changed since the COVID-19 pandemic.	4.38 ± 0.80	4.27 ± 0.79	4.25 ± 0.87	0.524	
My work has changed since the COVID-19 pandemic.	4.27 ± 0.74	4.30 ± 0.75	4.50 ± 0.70	0.800	
**Work Burden**					
The long time to confirm the result of the COVID-19 PCR test	4.47 ± 0.70	4.39 ± 0.79	4.39 ± 0.80	0.369	
Increased fatigue due to wearing protective clothing during work	4.36 ± 0.82	4.52 ± 0.66	4.56 ± 0.65	0.066	
Delay in patient treatment associated with the COVID-19 pandemic	4.47 ± 0.69	4.39 ± 0.70	4.44 ± 0.65	0.854	
Treatment process is different from usual, and the working process is not familiar.	4.17 ± 0.74	4.15 ± 0.81	4.08 ± 0.87	0.085	
Febrile, respiratory patient treatment guidelines are absent or frequently changed.	4.23 ± 0.70	4.28 ± 0.75	4.36 ± 0.72	0.361	
Risk of infection from contact with confirmed COVID-19 patients	4.38 ± 0.68	4.44 ± 0.70	4.50 ± 0.61	0.433	
**Lack of Resources**					
Lack of beds for febrile, respiratory patients	4.45 ± 0.64	4.42 ± 0.62	4.50 ± 0.61	0.820	
Lack of personal protective equipment	3.90 ± 1.02	3.74 ± 1.00	4.14 ± 0.90	0.660	2 < 3
Lack of medical staff and assistants due to the operation of the COVID-19 care unit.	4.39 ± 0.73	4.49 ± 0.71	4.58 ± 0.69	0.717	
**Personal Effect**					
I make more effort to maintain a healthy lifestyle after the COVID-19 pandemic.	3.21 ± 1.11	3.51 ± 1.08	3.47 ± 1.23	0.471	
I used to be worried about COVID-19-like symptoms during work.	3.60 ± 1.16	3.58 ± 1.17	4.00 ± 1.07	0.326	1,2 < 3
Working as frontline medical personnel around infectious patients has affected my family life.	4.02 ± 0.85	4.11 ± 0.90	4.28 ± 0.94	0.368	
I think I could be a threat to the safety of my family	4.37 ± 0.71	4.44 ± 0.59	4.56 ± 0.81	0.377	
Being frontline medical personnel around infectious patients has negatively affected my social and daily life.	3.38 ± 1.08	3.40 ± 1.14	3.81 ± 1.12	0.569	2 < 3

* *p* < 0.05, All items were measured on a 5-point Likert scale (“1-strongly disagree”, “2-disagree”, “3-neither agree nor disagree”, “4-agree”, “5-strongly agree”).

**Table 6 medicina-57-00583-t006:** Comparison of Likert Score According to Work Department.

	Likert Scale (Mean ± SD)			*p*	
	ED ^†^ (*n* = 263)	ICU ^‡^ (*n* = 21)	GW ^§^ (*n* = 23)		
**General**					
The experience of COVID-19 will enable us to cope well with new infectious diseases in the future.	3.92 ± 0.85	4.14 ± 0.73	3.96 ± 0.56	0.041 *	
COVID-19 pandemic is serious enough to change the emergency medical service system in the future.	4.35 ± 0.69	4.62 ± 0.59	4.74 ± 0.45	0.016 *	ED < GW
The information of each hospital’s bed status or closure is available in real-time.	3.33 ± 1.03	3.48 ± 0.98	3.22 ± 1.09	0.985	
The disaster training is helpful in the COVID-19 pandemic response	3.10 ± 1.05	4.17 ± 0.75	4.00 ± 0.82	0.394	ED < ICU, GW
**Changes in Work**					
The duration of my work has got longer since the COVID-19 pandemic.	3.62 ± 1.10	4.00 ± 0.78	3.65 ± 1.03	0.003 *	
The intensity of my work has changed since the COVID-19 pandemic.	4.29 ± 0.82	4.57 ± 0.51	4.26 ± 0.69	0.027 *	
My work has changed since the COVID-19 pandemic.	4.27 ± 0.76	4.48 ± 0.68	4.65 ± 0.49	0.198	
**Work Burden**					
The long time to confirm the result of the COVID-19 PCR test	4.43 ± 0.75	4.38 ± 0.67	4.17 ± 0.94	0.171	
Increased fatigue due to wearing protective clothing during work	4.45 ± 0.73	4.71 ± 0.56	4.57 ± 0.51	0.009 *	
Delay in patient treatment associated with the COVID-19 pandemic	4.40 ± 0.70	4.71 ± 0.46	4.39 ± 0.72	0.009 *	
Treatment process is different from usual, and the working process is not familiar.	4.10 ± 0.81	4.52 ± 0.60	4.43 ± 0.60	0.380	ED < ICU, GW
Febrile, respiratory patient treatment guidelines are absent or frequently changed.	4.29 ± 0.73	4.33 ± 0.66	4.39 ± 0.84	0.587	
Risk of infection from contact with confirmed COVID-19 patients	4.38 ± 0.69	4.81 ± 0.40	4.57 ± 0.59	0.000 *	ED < ICU
**Lack of Resources**					
Lack of beds for febrile, respiratory patients	4.46 ± 0.62	4.19 ± 0.68	4.39 ± 0.50	0.261	
Lack of personal protective equipment	3.74 ± 1.01	4.62 ± 0.50	4.17 ± 0.94	0.002 *	ED < ICU
Lack of medical staff and assistants due to the operation of the COVID-19 care unit.	4.49 ± 0.70	4.43 ± 0.81	4.35 ± 0.76	0.645	
**Personal Effect**					
I make more effort to maintain a healthy lifestyle after the COVID-19 pandemic.	3.34 ± 1.10	3.76 ± 1.10	3.91 ± 1.08	0.974	
I used to be worried about COVID-19-like symptoms during work.	3.55 ± 1.15	4.10 ± 1.26	4.13 ± 0.92	0.188	
Working as frontline medical personnel has affected my family life.	4.06 ± 0.91	4.48 ± 0.68	4.22 ± 0.74	0.659	
I think I could be a threat to the safety of my family	4.41 ± 0.67	4.71 ± 0.46	4.48 ± 0.59	0.050	
Being frontline medical personnel has negatively affected my social and daily life.	3.40 ± 1.11	4.05 ± 1.12	3.70 ± 1.19	0.851	ED < ICU

* *p* <0.05, ^†^ ED; Emergency Department, ^‡^ ICU; Intensive Care Unit, ^§^ GW; General Ward. All items were measured on a 5-point Likert scale (“1-strongly disagree”, “2-disagree”, “3-neither agree nor disagree”, “4-agree”, “5-strongly agree”).

## Data Availability

The data presented in this study are available on request from the corresponding author. The data are not publicly available due to privacy.

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
