# Peer review of "Comparison of COVID-19 Pandemic-Related Stress among Frontline Medical Personnel in Daegu City, Korea"

_medicina, 2021, doi:10.3390/medicina57060583_

Round 1
Reviewer 1 Report
Congratulations to the authors for the work presented, I just would like you to clarify some questions about the work presented. 1. Because they chose the general health questionnaire-12 (GHQ-12 on the existence of other questionnaires, with greater dissemination in scientific publications. 2. The calculation of the sample for the research, as well as the total populations of professionals of the different categories, would be a piece of information to know the impact of their work. 3. The results section is very extensive, it should highlight the most important findings. 4. The discussion should contain a greater number of citations and follow the order of the results, providing contrast between articles present in the current scientific literature, taking into account the level of research currently being carried out on Covid. 5. They have evaluated the possibility of updating the bibliography with citations of less than 5 years. With a simple search on any search engine (WOS or Pubmed), most of your citations older than 5 years could be updated.Author Response
Response to Reviewer 1 Comments
Point 1. Because they chose the general health questionnaire-12 (GHQ-12) on the existence of other questionnaires, with greater dissemination in scientific publications.
Response 1. Thank you for your comment. We considered many survey tools which can measure respondent’s stress before we conducted it, but at that time, our survey was conducted with other surveys, so the simplest and least time-consuming tool was selected for the respondents. If given the next opportunity, we will select a more scientifically proven survey tool and proceed with the research.
Point 2. The calculation of the sample for the research, as well as the total populations of professionals of the different categories, would be a piece of information to know the impact of their work.
Response 2. Daegu city has 16 emergency medical centers, 2 centers are level 1 ED, 4 centers are level 2, and 10 centers are level 3. The numbers of medical staff of each EDs are 149, 214, and 151 in each (as of March 2020). The number of frontline medical staff who care of COVID-19 patients in general ward and in ICU is about 150(as of March 2020, doctors 18, nurses 132, It changed frequently over time). With 141 doctors and 523 nurses, we conducted this survey, and among them, 307 (46.2%) responded. The number of frontline medical staff changes frequently, so we could not describe definitely in our manuscript. According to your comment, we revised and clarify the sample of the research. Thank you for your valuable comment.
Point 3. The results section is very extensive, it should highlight the most important findings.
Response 3. Thank you for your comment. We know that it is a little difficult to look at a glance to get to know the result of our survey, because it has many questions and there are many numbers to see the result. According to your comment, we update the ‘RESULT’ part concisely for highlighting the most important findings.
Point 4. The discussion should contain a greater number of citations and follow the order of the results, providing contrast between articles present in the current scientific literature, taking into account the level of research currently being carried out on Covid.
Response 4. Thank you for your valuable comment. We reviewed our manuscript, especially ‘DISCUSSION’ part, and we checked the order of description according to results and then updated references with taking into account the level of research currently being carried out on COVID-19.
Point 5. They have evaluated the possibility of updating the bibliography with citations of less than 5 years. With a simple search on any search engine (WOS or Pubmed), most of your citations older than 5 years could be updated.
Response 5. Thank you for your comment, we updated the bibliography with citrations of less than 5 years except SARS related citrations and firstly published paper about GHQ-12.

Reviewer 2 Report
Thank you for the opportunity to review the presented manuscript entitled “Comparison of COVID-19 pandemic-related stress among frontline medical personnel in Daegu city, Korea”. This study was aimed to evaluate the level and kind of stress and its relationship with job types, work departments, and medical centers among COVID-19 pandemic frontline medical staff. It should be emphasized that the topic addressed is timely and worthy of analysis. The results derived are interesting, however, to predict... The major limitation is the very local coverage, and the lack of follow-up analysis, likewise standardized questionnaires could have been used. Nevertheless, once corrections will be made, it is worth publishing the results of this study for further analysis. Apart from the mentioned limitations (which cannot be changed at this stage of the study), I have pointed out a few more minor observations.
Minor:
- The introduction is quite modest, I would suggest including a bit more information about medical procedures related to the treatment of patients. The point is to indicate how the rigor, the facilities of the medics' work differs in the context of COVID vs. pre-pandemic labor.
- Sample size calculation was not provided.
- Was the author's survey validated?
- Nothing was mentioned about the distribution of the data while indicating the use of parametric tests - needs clarification
- I have trouble understanding Table 3. First, I would point out in Table 3 in the footer a reminder of what the direction of the value is (which means a higher score). Secondly, in the description of the methods section, it states that a scale of 0-3 was used (line 89) and in the table, there are scores above 4, I don't understand why?
- Please correct editorial errors. Some paragraphs begin with a larger tab, some with a smaller tab (or space)
- Please indicate the limits of your study.
Author Response
Response to Reviewer 2 Comments
The major limitation is the very local coverage, and the lack of follow-up analysis, likewise standardized questionnaires could have been used. Nevertheless, once corrections will be made, it is worth publishing the results of this study for further analysis. Apart from the mentioned limitations (which cannot be changed at this stage of the study), I have pointed out a few more minor observations.
Point 1. The introduction is quite modest, I would suggest including a bit more information about medical procedures related to the treatment of patients. The point is to indicate how the rigor, the facilities of the medics' work differs in the context of COVID vs. pre-pandemic labor.
Response 1. Thank you for your comment. According to your comment, we updated our manuscript based on your opinion. For giving information about medical procedures related to the treatment of patients during COVID pandemic and pre-pandemic.
Point 2. Sample size calculation was not provided.
Response 2. Daegu city has 16 emergency medical centers, 2 centers are level 1 ED, 4 centers are level 2, and 10 centers are level 3. The numbers of medical staff of each EDs are 149, 214, and 151 in each (as of March 2020). The number of frontline medical staff who care of COVID-19 patients in general ward and in ICU is about 150(as of March 2020, doctors 18, nurses 132, It changed frequently over time). With 141 doctors and 523 nurses, we conducted this survey, and among them, 307 (46.2%) responded. The number of frontline medical staff changes frequently, so we could not describe definitely in our manuscript. According to your comment, we revised and clarify the sample of the research. Thank you for your valuable comment.
Point 3. Was the author's survey validated?
Response 3. Thank you for your valuable comment. Exploratory factor analysis (EFA) was carried out to determine factorial validity of our questionnaire. Prior to extraction of factors, Kaiser-Meyer-Olkin (KMO) measure of sampling adequacy was checked to evaluate the fitness of the data for factor analysis. Initially, Principal component analysis with Varimax rotation was used to extract factors based on multiple criteria, including Eigenvalue >1, the Scree test, factor loading coefficient > 0.4 and the cumulative percent of variance extracted. Finally, parallel analysis was carried out to determine the ultimate factors to be related. The internal consistency reliability of each factorially derived scale was assessed by calculating Cronbach’s alpha. The KMO measures of sampling adequacy value was 0.848, exceeding the recommended value of 0.6 and Bartlett’s test of sphericity reached statistical significance (p=0.000), supporting factorability of the correlation matrix. PCA with Varimax rotation demonstrated the presence of four components with Eigenvalues exceeding one. An inspection of the Scree plot also revealed a clear break after the 4th component. Using this result, it was decided to retain four component solutions, which explained a total of 62.8% of the variance. And we added ‘General’ component which consists of four questions. We defined these five components as ‘General’, ‘Changes in Work’, ‘Work Burden’, ‘Lack of Resources’, and ‘Personal Effect’ respectively. Cronbach’s alpha ranged from 0.655 for the scale ‘General’ to 0.766 for the scale ‘Work Burden’. According to your opinion, we updated our manuscript.
Point 4 Nothing was mentioned about the distribution of the data while indicating the use of parametric tests - needs clarification
Response 4. Thank you for your comment. We described the way of data analysis in ‘MATERIALS AND METHOD’. Using IBM SPSS, we conduct Chi-square test, t-test and ANOVA as well as frequencies and percentage for nominal variables and means and standard deviation for continuous variables. For continuous variables (eg. Comparison of Likert scale between groups), comparisons were performed using ANOVA, and detailed comparisons were performed through post-hoc analysis. We updated our manuscript.
Point 5 I have trouble understanding Table 3. First, I would point out in Table 3 in the footer a reminder of what the direction of the value is (which means a higher score). Secondly, in the description of the methods section, it states that a scale of 0-3 was used (line 89) and in the table, there are scores above 4, I don't understand why?
Response 5. According to your first comment, we insert the footer which reminds of the direction of the value. And for your second comment, we described the scale of our questionnaire in line 88-91. GHQ-12 has a scale of 0-3, and other questions have a scale of 1-5. We insert the footer for clarifying it in each table. Thank you for your comment.
Point 6. Please correct editorial errors. Some paragraphs begin with a larger tab, some with a smaller tab (or space)
Response 6. We reviewed our manuscript again and again, we correct some or editorial errors, thank you for your comment.
Point 7. Please indicate the limits of your study.
Response 7. Thank you for your comment. In the revision process, we found the weakness of our manuscript once again. As you mentioned above, we supposed to describe limitations of our study in ‘DISCUSSION’. Based on your comments, we updated our manuscript. Thank you again for your valuable opinion.
